# Ultrasound Assessment of the Median Nerve Does Not Adequately Discriminate the Carpal Tunnel Syndrome among Patients Diagnosed with Diabetes

**Carlos Antonio Guillen-Astete [1,\*], Monica Luque-Alarcon [2] and Nuria Garcia-Montes [3]**

1    Rheumatology Department, Ramón y Cajal University Hospital, 28034 Madrid, Spain
2    Faculty of Medicine, Autónoma University of Madrid, 28049 Madrid, Spain; millu99@hotmail.com
3    Emergency Department, Ramón y Cajal University Hospital, 28034 Madrid, Spain; gmontesnuria@gmail.com
\*    Correspondence: carlos.guillen@salud.madrid.org

**Abstract:** Background: Carpal tunnel syndrome is the most prevalent peripheral nerve entrapment condition of the upper limb. Among metabolic risk factors, diabetes is considered the most relevant. Although wrist ultrasound assessment of the median nerve has demonstrated a good correlation with the gold standard for the diagnosis of this syndrome, neurophysiological study, its usefulness in patients with diabetes is questionable because the compressive phenomenon is not the predominant one. Method: We conducted a retrospective study to compare the clinical and median nerve ultrasound features of patients with carpal tunnel syndrome previously diagnosed or not diagnosed with diabetes. Additionally, a linear multivariate regression analysis was performed to determine to what extent the cross-sectional area of the median nerve was dependent on the condition of diabetes by fixing other variables such as sex, age, or time of evolution. Results: We included 303 records of patients (mean age $44.3 \pm 11.7$ years old, 57.89% female, mean of time of evolution $13.6 \pm 8.3$ months) from 2012 to 2020. The cross-sectional area of the median nerve was $10.46 \pm 1.44$ mm$^2$ in non-diabetic patients and $8.92 \pm 0.9$ mm$^2$ in diabetic patients ($p < 0.001$). Additionally, diabetic patients had a shorter time of evolution ($7.91 \pm 8.28$ months vs. $14.36 \pm 0.526$ months, $p < 0.001$). In the multivariate analysis, the resultant model (fixed R-square = 0.659, $p = 0.003$) included a constant of the following four variables: the evolution time (Beta coeff. = 0.108, $p < 0.001$ 95% CI 0.091 to 0.126, standardized coeff. = 0.611), the condition of diabetes (Beta coeff. = $-0.623$, $p < 0.001$ 95% CI $-0.907$ to $-0.339$, standardized coeff. = $-0.152$), the severity (Beta coeff. = 0.359, $p = 0.001$ 95% CI 0.147 to 0.571, standardized coeff. = 0.169), and the masculine sex (Beta coeff. = 0.309, $p = 0.003$, 95% CI 0.109 to 0.509, standardized coeff. = 0.103). Conclusions: Ultrasound assessment of the median nerve in patients with diabetes is not a useful tool to confirm whether carpal tunnel syndrome should be diagnosed or not diagnosed.

**Keywords:** carpal tunnel syndrome; ultrasound; power doppler signal; diabetes

## 1. Introduction

Carpal tunnel syndrome (CTS) is the most prevalent peripheral nerve entrapment pathology. Its estimated annual incidence is 125 cases per 100,000 inhabitants [1]. It is 1.4 times more frequent in women than in men, and it is presumed that this increased risk is due to the higher prevalence of osteoarthritis in women [1–3]. However, other risk factors have been identified, such as diabetes and hypothyroidism [4]. Although CTS is pathophysiologically explained as a result of the continuous mechanical stress of the median nerve as it passes through the carpal tunnel, in the case of patients with diabetes, the origin of the symptoms is due to metabolic causes linked to peripheral neuropathy, rather than purely mechanical [5,6].

Recently, attempts have been made to evaluate the role of carpal ultrasound in the diagnosis of CTS [7–10]. In all cases, the initiative to validate ultrasound as a diagnostic

tool is justified by its greater availability and accessibility. However, since it is presumed that the biomechanical entrapment of the median nerve is not the main triggering factor of this disease in patients with diabetes, ultrasound evaluation in these patients would likely have a minor role.

The purpose of the present study is to determine the differences in the ultrasound examination of patients with CTS as a function of whether or not they were previously diagnosed with diabetes.

## 2. Method

A cross-sectional study of patients with an electrophysiological diagnosis of CTS based on available information from three databases was conducted between January and March 2021.

The records included patients diagnosed in three different centers in the Community of Madrid between 2012 and 2020. Only patients with at least one ultrasound examination of the carpus, including the cross-sectional area of the median nerve inside the carpal tunnel and detection of hyperemia using power Doppler (PD) signal, were included. Records of patients with thyroid disorders, osteoarthritis, amyloidosis, and connective tissue diseases were excluded. In addition, records corresponding to patients already treated due to CTS on the same hand were excluded.

All data were obtained from the corresponding electronic records. In the ultrasound examination, the cross-sectional area of the median nerve—measured in square millimeters—and the result of the detection of PD signal inside the carpal tunnel were extracted. Three different ultrasound devices were used (Logiq S9 General Electric®, Nemio XG Toshiba®, and MyLab 7 Esaote®); however, all the studies were performed by the same rheumatologist following the recommendations of Filippucci et al. for median nerve assessment [11]. Although it has been demonstrated that the place of measure along the carpal tunnel has no effect on the area of the median nerve [12], all measurements were performed at the level of the pisiform bone. In categorical, ordinal terms, disease severity was determined via neurophysiological evaluation and according to the definitions of the American Association of Electrodiagnostic Medicine [13]. The diagnosis of diabetes was only obtained directly from the medical chart when it was established at least 5 years ago.

Patients were grouped according to whether or not they were diagnosed with diabetes, and all other clinical, epidemiological, and ultrasound characteristics were compared. A multiple regression analysis was performed to correlate the median nerve area with the time of evolution and severity of CTS.

In order to determine clinical and epidemiological differences among CTS patients with or without a previous diagnosis of diabetes, we performed a bivariate analysis using chi-square and T-student tests (*p*-value significance fixed at 0.10). Additionally, to assess the relative weight of the prior diagnosis of diabetes among patients with CTS, a linear regression multivariate test was performed considering the section area of the median nerve as the dependent variable using the forward stepwise method of variable inclusion.

For purposes of multivariate analysis, female sex was categorized as 0 (male as 1), the severity of disease was categorized from 1 to 3 (mild to severe), dichotomic variables were categorized as present (1) and absent (0), and finally, treatment response was classified from 0 to 2 (none, partial and complete).

Our local scientific research ethics committee approved the conduct of the presen study.

## 3. Results

Three hundred and three records were included for analysis. Forty-seven patients (15.5%) had a diagnosis of diabetes. The mean age ± SD was 44.3 ± 11.7 years old. One hundred and seventy-five records (57.89%) corresponded to female patients. The distribution of severity of CTS according to the neurophysiological diagnosis study was as follows: 61 (20.1%) mild, 153 (50.5%) moderate, and 80 (29.4%) severe. The mean time of

evolution of CTS was 13.6 ± 8.3 months. Thirty-six patients (11.9%) had previously been diagnosed with CTS in the contralateral hand.

No significant differences in terms of age were detected in patients with or without diabetes; however, the cross-sectional area of the median nerve was 10.46 ± 1.44 mm$^2$ in non-diabetic patients and 8.92 ± 0.9 mm$^2$ in diabetic patients ($p < 0.001$). Intra tunnel power Doppler signal was detected in 12 non-diabetic patients (4.6%) and was not detected in non-diabetic patients. (Additionally, diabetic patients had a shorter time of evolution (7.91 ± 8.28 months vs. 14.36 ± 0.526 months, $p < 0.001$). Among diabetic patients, the antecedent of a previous contralateral CTS was present in 13 subjects (27.7%), while in non-diabetic patients, it was recorded in 23 (9.0%) ($p = 0.001$, OR 3.873; 95% CI 1.794 to 8.361). Table 1 shows the characteristics of the records included in the study, differentiating subjects according to their diabetic or non-diabetic patient condition.

**Table 1.** Demographic and clinical characteristics of the population of registries included in the study. *p*-value has been calculated for chi-square or Student's *t*-test as appropriate. CTS: carpal tunnel syndrome.

| Variable | Patients with Diabetes N = 47 | Non-Diabetic Patients N = 256 | *p*-Value |
|---|---|---|---|
| Age (years ± standard deviation) | 44.45 ± 11.25 | 44.2 ± 11.85 | 0.896 |
| Female sex (%) | 31 (65.9) | 144 (56.3) | 0.261 |
| Time of evolution (months) | 7.91 ± 5.67 | 14.36 ± 8.41 | <0.001 |
| Severity (%)<br>Mild<br>Moderate<br>Severe | 18 (38.3)<br>28 (59.6)<br>1 (2.1) | 43 (16.8)<br>125 (48.8)<br>88 (34.4) | <0.001 |
| Cross-sectional area of the median nerve (mm$^2$) | 8.92 ± 0.90 | 10.46 ± 1.44 | <0.001 |
| Previous contralateral CTS diagnosis (%) | 13 (27.7) | 23 (9.0) | 0.001 |
| Historic response to conservative treatment (splint) (%)<br>None<br>Partial<br>Complete | 37 (78.7)<br>9 (19.1)<br>1 (2.1) | 103 (40.2)<br>132 (51.6)<br>21 (8.2) | <0.001 |
| Historic response to corticoids local administration (%)<br>None<br>Partial<br>Complete | Patients treated = 46<br>35 (76.0)<br>6 (13.0)<br>5 (10.9) | Patients treated = 179<br>55 (30.7)<br>101 (56.4)<br>23 (12.8) | <0.001 |

In the bivariate analysis, Pearson correlation with the cross-sectional area of the median nerve was statistically significant with the time of evolution of the clinical manifestations (R = 0.782, $p < 0.001$). No other numerical variable showed a significant correlation with the cross-sectional area of the median nerve. Among categorical variables, besides the condition of diabetes, females had a smaller area than males (10.03 ± 1.57 vs. 10.48 ± 1.32 mm$^2$, $p = 0.007$).

In the linear regression multivariate analysis, the resultant model (fixed R-square = 0.659, $p = 0.003$ and ANOVA F-test = 147.231, $p < 0.001$) included a constant of 7.994 mm$^2$ and four variables: the evolution time, the condition and severity according to the neurophysiology study, and the masculine sex (Table 2). The predictive model including these four variables showed no significant differences with the real cross-sectional area of the median nerve (diff = −0.0041 ± 0.861, $p = 0.934$).

**Table 2.** Results of the linear multivariate regression analysis after a forward step-wise modeling. CTS: Carpal tunnel syndrome.

| Variable | Beta Coefficient | 95% Confidence Interval | *p*-Value | Standarized Coefficient |
|---|---|---|---|---|
| Evolution time | 0.108 | 0.091 to 0.126 | <0.001 | 0.611 |
| Diagnosis of diabetes | 0.623 | −0.907 to −0.339 | <0.001 | −0.152 |
| Severity of the CTS according to neurophysiology | 0.359 | 0.147 to 0.571 | 0.001 | 0.169 |
| Sex male | 0.309 | 0.109 to 0.509 | 0.003 | 0.103 |

No significative differences in the cross-sectional area of the median nerve were detected when compared patients with and without the diagnosis of diabetes, according to their level of severity (data not shown).

## 4. Discussion

According to our results, ultrasound examination of the median nerve as it passes through the carpal tunnel is of scarce diagnostic value in diabetic patients since the classic reference of the increase in the cross-sectional area of the nerve does not seem to take place in these patients. In our series, we also identified a higher proportion of relapsing patients and a lower response to conservative treatment with splints and infiltrations.

Our study has certain limitations that we feel are appropriate to discuss. First, the purpose of the study is limited to determining to what extent the condition of diabetes influences the clinical characteristics of the disease and the fundamental diagnostic value of ultrasound. The detection of PD signal was not comparatively analyzed due to the absence of cases in the group of diabetic patients. Furthermore, because this was a retrospective study with data from three different ultrasound devices, PD signal detection may have been heterogeneous.

Another limitation to highlight is the accuracy of determining the magnitude of the dependent variable of the linear regression analysis. Two ultrasound devices yielded absolute values, while the third was sensitive to one-tenth of a square millimeter.

Finally, the neurophysiological studies, although they used the same classification pattern as a reference, were performed in three different centers and not by the sameprofessional.

The cross-sectional area of the median nerve was proposed for use as a diagnostic tool that is easy to obtain and correlates well with the results of neurophysiological studies [9,14–16]. Although its determination is simple and more accessible than the electromyogram, it has been suggested that its reliability could be related to specific anthropometric characteristics [17]. This would imply that the cut-off points of normality should be adjusted to body mass index [18,19] or carpal circumference [17].

In our study, diabetic status was a contributory variable in the nerve thickness prediction model, albeit in a negative sense. This can be interpreted to mean that the genesis of clinical CTS in diabetic patients is not due to mechanical nerve injury but to peripheral neuropathy [6]. It also implies that once the clinical manifestation has developed, patients are diagnosed earlier and, therefore, with a lesser degree of median nerve involvement. The causal relationship between diabetes and CTS is not due to the classic mechanical entrapment syndrome. The absence of cross-sectional changes in the median nerve favors a non-mechanical cause. Recent studies point to the development of nerve fiber fibrosis mediated by Transforming Growth Factor (TGF-β), Vascular Endothelial Growth Factor (VEGF) and certain interleukins [20,21].

The lack of therapeutic response to splints or infiltrations supports the idea that the cardinal lesion of CTS in diabetic patients is not necessarily a repetitive microtrauma. However, as it has been previously suggested, it can have a triggering relationship [21–23].

The asymmetric distribution of CTS severity between patients with and without a previous diagnosis of diabetes also suggests that the disease in people with diabetes tends to be milder. However, the degree of severity was also linked to the time of evolution [4]. The time to diagnosis of CTS was 50% shorter in diabetic patients than in non-diabetics.

Early diagnosis could be due to the follow-up that diabetic patients have and their greater degree of alertness around neurological symptoms.

## 5. Conclusions

In diabetic patients, the determination of the cross-sectional area of the median nerve via ultrasound should be used as a single discrimination tool for diagnostic purposes. This lack of diagnostic validity may be due to the earliness of the diagnosis, the lack of cardinal mechanical lesion, or the interaction of both considerations.

Thus, in diabetic patients, confirmation of the diagnosis of CTS, once a suspicious clinical picture has been established, should be made through neurophysiological studies.

**Author Contributions:** Conceptualization, C.A.G.-A. and M.L.-A.; methodology, C.A.G.-A.; software, C.A.G.-A.; validation, C.A.G.-A., M.L.-A. and N.G.-M.; formal analysis, C.A.G.-A., M.L.-A. and N.G.-M.; investigation, C.A.G.-A., M.L.-A. and N.G.-M.; resources, N.G.-M.; data curation, N.G.-M.; writing—original draft preparation, C.A.G.-A.; writing—review and editing, C.A.G.-A.; visualization, C.A.G.-A.; supervision, C.A.G.-A.; project administration, C.A.G.-A.; funding acquisition, none. All authors have read and agreed to the published version of the manuscript.

**Funding:** This research received no external funding.

**Institutional Review Board Statement:** The study was conducted according to the guidelines of the Declaration of Helsinki, and approved by the Ethics Committee of our institution (protocol code TFG033-20_HUQM, March 2020).

**Informed Consent Statement:** Patient consent was waived due to not patient was personaly recruited for purposes of this study. All the information used was obtained from electronic medical registries.

**Data Availability Statement:** The data set of the present study is available in Synapse™ repository (ID: syn26338284).

**Conflicts of Interest:** The authors declare no conflict of interest.

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
