# Peer review of "Ultrasound Assessment of the Median Nerve Does Not Adequately Discriminate the Carpal Tunnel Syndrome among Patients Diagnosed with Diabetes"

_diabetology, doi:10.3390/diabetology2040020_

Round 1

Reviewer 1 Report

Many thanks for the opportunity to review this important and interesting work. I enjoyed reading the article very much and hope that the below comments help to improve your manuscript.

  1. There are many evidence synthesis articles defining the US CSA thresholds for the diagnosis of CTS which are not cited (e.g. 10.1016/j.apmr.2017.08.489, 10.1097/PHM.0000000000001104, 10.1111/ene.14759) - given they represent the 'best evidence' they could be added to the intro/discussion & their findings discussed in the context of your work on diabetes?
  2. When taking the CSA of the median nerve, was the epineurium included? And what location was the measurement taken from, please? Was it the same location for each patient (e.g. level of the pisiform or otherwise)?
  3. In accordance with the STROBE and SAMPL guidance, please can you explain how and why (perhaps even with DAGs) you selected the covariables that you did (time of evolution, diabetes, severity and sex) for your multivariable model. I see that you described "to assess the relative weight of the prior diagnosis of diabetes among patients with CTS, a linear regression multivariate test was performed considering the section area of the median nerve as the dependant variable using the stepwise method of variable inclusion." but this doesn't make sense to me - did you put all your available covariables in and allow the stats package to exclude non-significant variables in a backwards step-wise fashion? Please expand your methods sufficiently to enable replication
  4. When describing means and their variance (such as p3, line 95) please define what the variance statistic is (e.g. "13.6 (SD 8.3) months" if 8.3 is the standard deviation) rather than "13.6 ± 8.3 months" because in the present form, it's unclear whether 8.3 is the standard deviation, standard error, IQR, some otherwise... Same again for p3 lines 98 and 99 for the CSA measurements.
  5. Please move lines 113-122 into a table, such that readers can see the univariable as well as multivariable co-efficients for each of the covariables you explored.
  6. Figure 1 can be removed as it adds nothing to the results. 
  7. Could you create some artwork to show your findings? Perhaps what 'normally' happens to CSA in CTS vs what happens to diabetic patients, so far as you've observed? 
  8. You mentioned that neurophysiological data were captured but why is this not presented? It would be valuable to know the associations between EDX values, symptoms, CSA, etc
  9. Although I am not a proponent of US CSA (and I do agree with the authors' assertions), I feel that the conclusion "In diabetic patients, ultrasound determination of the cross-sectional area of the median nerve is of no diagnostic value." is too strong. There are several unresolved issues in CTS diagnostics broadly speaking, this study lacks a reference standard, CSA may add some value as part of a multimodal assessment (e.g. a cocktail of clinical assessments, EDC, US, dMRI, tomography) and so this definitive statement seems too harsh. Consider toning down? 
  10. In line with widespread calls, good practice, and transparency, the authors should publish their anonymised dataset (spreadsheet of data) and all available medical images (e.g. DICOMs of the US examination) open-source, either with the manuscript on the publishers website or better still, in a public repository e.g. OSF. This permits IPD meta-analysis, 3rd party modeling of the data, development of software/pipelines to better analyse medical image data, etc.

With best wishes,

Ryckie Wade

NIHR Doctoral Research Fellow and Specialty Training Registrar Graduate Statistician, Royal Statistical Society
Associate Lead for Plastic Surgery, Royal College of Surgeons of England  The National Hand Registry Committee, British Society for Surgery of the Hand 

Author Response

  1. All the references you suggested have been included in the manuscript in the discussion and introduction sections.
  2. All the measurements were performed at the level of the pisiform bone.
  3. The number of registries included in our database is 300+. According to the formal modeling of multivariate regression, we were able to include up to 30 variables, much more than selected by relevance according to the studies consulted and those variables that were eligible in the bivariate analysis.  Considering that our model is mainly explicative rather than predictive, we chose a forward step-wise method. We have clarified this aspect in the method section.
  4. Thank you for your observation. Indeed, it is the SD. It is well remarked into the text in the first appearance of the "±" in the results section.
  5. A new table has been added to the manuscript, including the information suggested by the reviewer.
  6. As you recommended, figure 2 has been removed.
  7. Thank you for your request. We find it interesting; however, this observational study was not designed to explore causalities but to test a hypothesis founded in our experience.
  8.  Thank you for your request. The main objective of our study was to determine if ultrasound studies can discriminate a CTS in patients with diabetes. That consideration leaves the neurophysiological studies in a second-place of relevance. Also, in our clinical setting, neurophysiological studies are performed in different places using technics and equipment that are not the same even in the same unit. Those reasons prevent us from considering the separate analysis of the neurophysiological studies.
  9. Your recommendation is reasonable. We have modulated our conclusions. 
  10. Our data set has been uploaded to Synapse data repository. The reference is now shown in the manuscript. As we stated in the method section. Data about the measures of the median nerve was performed by a single rheumatologist who informed into the corresponding medical record. Very few images were available for the present study.

Reviewer 2 Report

The authors argued that CTS hands with diabetes showed different pathophysiology than non-diabetic CTS hands, which was confirmed by differences in ultrasound findings. This is an interesting topic.

Abstract

  1. Abstracts should be concise. In particular, it is better to avoid mentioning etiologies that are not related to the main finding of this study in the ‘Background’ section.

Materials and Methods

  1. Line 62: Please specify ‘metabolic or endocrinologic diseases’ you excluded. The authors argue that metabolic causes contributed to the difference in CSA between groups. If so, there should be a clear explanation of which metabolic or endocrinologic diseases other than diabetes were considered and excluded.

  1. Line 74: How did you determine whether the patient had diabetes? Was it through simple history taking?, taking blood glucose lowering agents or not?, or though laboratory findings? Diabetes is the main variable of this study. Would you please explain it clearly?

  1. Line 84: It is necessary to move this ethical statement, which suddenly appears while explaining the statistical method, to another place.

Results

  1. CTS is a syndrome with a vast spectrum depending on its severity rather than being considered as a single disease entity. It has already been shown that neurophysiologic pathologies also differ according to the degree and progress of peripheral nerve injury (PMID: 16009539, 22989216). Therefore, it is important to diagnose CTS and look at the differences over the course of the disease. According to the study design, it seems that you can suggest the difference in CSA according to CTS severity from your dataset. For example, compare the difference in CSA between diabetic and non-diabetic groups on EDx-mild grade. I recommend that you add these kinds of results and discuss them. That would be able to strengthen your findings.

Discussion

  1. Although the authors noted this as a limitation, it would be worth discussing the experimental or clinical evidence of what metabolic causes might have been involved. For the sake of the flow of this study, this cannot be omitted. A more thorough literature review is needed.

  1. Line 170-171: Despite the results of this study, ultrasonography is still the main diagnostic tool of CTS and believed that it provides a lot of information. I can't entirely agree with your conclusion that ultrasonography is of no diagnostic value for CTS with diabetes. Also, your results do not sufficiently support this conclusion. In the same context, the conclusion part of the Abstract also needs to be revised.

Author Response

  1. Thank you for your observation. Changes have been made in the abstract.
  2. Disorders have been detailed.
  3. The diagnosis of diabetes was consigned directly from the medical chart (registered by a physician) and only when it was established at least five years ago. These data have been included in the method section.
  4. The ethical topic has been relocated at the end of the method section. 
  5. Thank you for your kind suggestion. The main interest of our study is to determine the usefulness of US discriminating CTS in patients with diabetes besides its severity. We have followed your recommendation and perform and stratified analysis in the three groups of severity. No significant results were found. Reference to these findings has been included in the manuscript.
  6. Agree. Although our main objective was not linked to assess an etiology, we have included some hypothesis that justifies the absence of morphological changes in the CSA of the nerve and the CTS.
  7. After a careful analysis of our results, we agree with this recommendation. We have modulated our conclusión.

Round 2

Reviewer 2 Report

The manuscript has been improved. 

Thank you for the author's efforts.